# Autonomous Underwater Vehicle Path Planning Method of Soft Actor–Critic Based on Game Training

Zhuo Wang, Hao Lu , Hongde Qin * and Yancheng Sui

School of Naval Engineering, Harbin Engineering University, Harbin 150001, China
* Correspondence: qinhongde@hrbeu.edu.cn

**Abstract:** This study aims to solve the issue of the safe navigation of autonomous underwater vehicles (AUVs) in an unknown underwater environment. AUV will encounter canyons, rocks, reefs, fish, and underwater vehicles that threaten its safety during underwater navigation. A game-based soft actor–critic (GSAC) path planning method is proposed in this study to improve the adaptive capability of autonomous planning and the reliability of obstacle avoidance in the unknown underwater environment. Considering the influence of the simulation environment, the obstacles in the simulation environment are regarded as agents and play a zero-sum game with the AUV. The zero-sum game problem is solved by improving the strategy of AUV and obstacles, so that the simulation environment evolves intelligently with the AUV path planning strategy. The proposed method increases the complexity and diversity of the simulation environment, enables AUV to train in a variable environment specific to its strategy, and improves the adaptability and convergence speed of AUV in unknown underwater environments. Finally, the Python language is applied to write an unknown underwater simulation environment for the AUV simulation testing. GSAC can guide the AUV to the target point in the unknown underwater environment while avoiding large and small static obstacles, canyons, and small dynamic obstacles. Compared with the soft actor–critic(SAC) and the deep Q-network (DQN) algorithm, GSAC has better adaptability and convergence speed in the unknown underwater environment. The experiments verifies that GSAC has faster convergence, better stability, and robustness in unknown underwater environments.

**Keywords:** autonomous underwater vehicle; optimal path planning; deep reinforcement learning; unknown underwater environment; particle swarm optimization

## 1. Introduction

Autonomous underwater vehicles have gained growing attention due to their irreplaceable role in marine data collection, subsea pipeline repair, subsea oil exploration, working in collaboration with divers, and thermocline analysis [1–7]. Path planning and dynamic obstacle avoidance enable AUVs to reach the mission target safely without colliding with obstacles, which is an important guarantee for AUVs to perform the mission, and is the core technology of AUV autonomy. Path planning is divided into global path planning of known electronic charts and local path planning with unknown obstacles. The ability of AUV local path planning in a dynamic or uncertain underwater environment is crucial [8]. In recent years, a variety of local path planning methods have been proposed. These methods improve the autonomy of AUVs, which mainly includes traditional methods and methods with learning capability.

The traditional local path planning methods mainly include Rapidly-exploring Random Tree (RRT), Artificial Potential Field (APF), and Fuzzy Logic Algorithm. It enables the AUV to avoid static and dynamic obstacles, and its path planning methods designed for specific working conditions is effective. However, these methods need to design the parameters of the algorithm according to underwater conditions, and the design of the algorithm depends more on the designer's understanding of the underwater environment.

Moreover, these methods do not have a learning capability and cannot improve the AUV path planning capability with task execution, so their performance is limited by the designer's parameter design level. Traditional local path planning methods are listed below. Li et al. [9] proposed an automatic ground map building and path planning algorithm in unmanned aerial/ground vehicles' (UAV/UGV) cooperative systems, which outperforms the genetic algorithm and A star algorithm in path cost. Hermand et al. [10] proposed a constrained control scheme based on the Explicit Reference Governor framework. The experiment verified that this method can control UAV while avoiding obstacles in the laboratory environment. Nie et al. [11] proposed an improved RRT algorithm, which simplifies the environment model by simplifying the representation of space, thus avoiding the dimensional catastrophe during computation and improving the computational speed. Li et al. [12] combined rolling planning with node screening and applied the improved RRT algorithm to underwater search and interception. Zacchini et al. [13] used RRT for path planning and realized the submarine terrain inspection using forward-looking sonar. Franco et al. [14] employed the APF algorithm to achieve the obstacle avoidance of AUV using scanning sonar. Noguchi et al. [15] employed the APF algorithm to implement an intervention of the autonomous underwater vehicle (I-AUV) in the process of catching sea urchins. Considered the dynamic constraints of the UAV, Tang et al. [16] proposed a trajectory planning algorithm based on the minimum snap trajectory method. The simulation experiment verified that this method can optimize the time and length of the generated trajectory in a simulation environment with simplified quay crane model. Meng et al. [17] proposed the prediction planning interception (PPI) algorithm based on the APF algorithm. This method determines the interception position by the motion tracking of the target and employs the APF algorithm to plan the interception route, so as to achieve the moving target interception in the ocean current environment of the harbor. Fan et al. [18] proposed an improved APF algorithm, and added a distance correction factor to the exclusion function to solve the local minimum problem. Lin et al. [19] realized the path planning of multi AUVs through the APF method by considering potential underwater obstacles and AUV dynamics. Li et al. [20] employed an improved APF algorithm to achieve obstacle avoidance in the process of target tracking. Li et al. [21] designed a 3-input controller based on the fuzzy logic algorithm with obstacle distance change as input, which can achieve obstacle avoidance in the same direction as AUV. Traditional local path planning algorithms achieve obstacle avoidance of AUVs for both static and dynamic obstacles. However, these methods need to design the algorithm for different working conditions, and the performance of the algorithm is constrained by the experience of the designer. The algorithm does not have a learning capability and is easily trapped in local minima. Therefore, an algorithm with learning capabilities is subsequently proposed.

The Reinforcement Learning (RL)-based path planning method is a typical algorithm with learning capability. Distinguishing from traditional methods, the RL-based local path planning method does not need guidance signal in the unknown environment, and adapts to the environment by online learning and continuous trial and error. Reinforcement learning enables the AUV to gradually adapt to the environment and make decisions through training. It has a good generalization ability, and is suitable for complex and variable application scenarios. Li et al. [22] combined the heuristic search strategy with Q-learning to reduce the energy consumption of mobile robot paths. Duguleana et al. [23] implemented the path planning of mobile robots with Q-learning. This method realizes the obstacle avoidance of moving obstacles when global information is available. Taghavifar et al. [24] combined the chaotic metaheuristic optimization with Q-learning to solve the obstacle avoidance path planning for robots under mobile obstacles. Singla et al. [25] combined the DQN method. Ref. [26] with UAV to achieve an obstacle avoidance of UAV in an unknown indoor environment. Sun et al. [27] combined the hierarchical deep Q network (HDQN) with a prioritized experience replay to propose an AUV path planning method for 3D ocean conditions. The path planning was divided into three layers to reduce the dimensionality of the path planning task and avoid dimensional disaster. Simulation experiments and

realistic tests verified that the algorithm can reduce the reinforcement learning training time and is safe and effective. Zhang et al. [28] proposed the deep interactive RL method by adding human rewards on the basis of DQN. This method possesses faster convergence and better performance than DQN. Yuan et al. [29] proposed an improved method based on the double deep Q-network (double-DQN) method [30], which outperforms the double-DQN algorithm in terms of the success rate and obstacle avoidance performance. The RL-based local path planning method can perform obstacle avoidance in unknown or dynamic environments without modeling the environment. Pei et al. [31] proposed the Dyna-Q algorithm by combined Q-learning based on Dyna architecture with simulated annealing mechanism, heuristic search strategies, and the reactive navigation principle. The practical experiment conducted by MATLAB and the robot operating system verified that this method can fulfill autonomous navigation tasks in the real world. Cui et al. [32] proposed a multi-layer Q-learning path planning method. The method used a two-layer structure to handle local global information. Simulation experiments demonstrate the effectiveness of this path planning method. Considering the policy selection in the early search process of Q-learning, Ma et al. [33] proposed the continuous local search Q-Learning (CLSQL) method. In this method, global environment was divided into independent local environments. The search between each intermediate point in the local environment was realized to reach the destination. This method outperforms Q-Learning, SARSA($\lambda$), and DQN in convergence speed and computation time. Focused on the slow convergence speed of UAV path planning, Boming et al. [34] proposed the guided Sarsa algorithm. This method enhanced the convergence speed due to the return function based on the position information and improved status update strategy. Khamidehi and Sousa [35] combined the double-DQN with a graph-based global path planning algorithm. This method improved the safety of the UAV path planning in a dynamic environment. Cao et al. [36] proposed a new asynchronous advantages actor–critic (A3C) method [37] based on the asynchronous variant of the actor–critic, which completes the target search in the simulation environment. Biferale et al. [38] focused on the path planning in a ship sailing in a 2D turbulent sea and implemented the actor–critic (AC) method [39] for ship path planning. Sun et al. [40] proposed the Sum Tree-DDPG method based on the deep deterministic policy gradient (DDPG) [41] method. This method improves the replay memory in the DDPG method and sets the reward function with reference to the APF method, which increases the stability of the AUV path planning method in an underwater canyons environment. Hong et al. [42] combined the twin-delayed deep deterministic policy gradient (TD3) method [43] with the frame stacking technique. This method can reduce the energy consumption of drones with the global energy-efficient path. Lan et al. [44] improved the DDPG algorithm to solve the path planning of the underwater glider (UG) in an ocean current environment. This method integrated the UG kinematic motion into MDP in DDPG. The simulation experiment verified that this method can fulfill the UG autonomous navigation tasks in the ocean current environment based on the Tokyo Bay geography and the unacquainted ocean. The RL algorithm requires a lot of exploration to adapt to the environment, and therefore has high demands on the environment.

Due to the unknown and complex underwater environment, AUV needs to be explored in various environments. The high risk and cost of AUV field experiments determine that the simulation environment needs to be constructed to train AUV. The obstacle parameters (size, shape, number and location) in the simulation environment are generally designed based on human experience, so it is difficult to guarantee the exploration of RL. The game theory is introduced to SAC to meet the requirement of exploration in the simulation environment.

The study is organized, as follows: in Section 2, the framework of the AUV path planning method is introduced and the mathematical model of the underwater environment is defined. The components of the AUV path planning method are presented in Section 3. In Section 4, the AUV path planning and obstacle avoidance method are discussed for path planning experiments in an unknown simulation environment. Section 5 draws the conclusions.

## 2. Materials and Methods

To enhance the exploration capacity and training speed, Haarnoja et al. [45] proposed the SAC method. Compared with other RL methods, the SAC adds action entropy to ensure the exploration ability of agent strategies and improve the randomness of agent actions. To solve the problem of insufficient diversity of the simulation environment, we introduce game theory into the training of SAC and propose game-based SAC (GSAC) path planning method. GSAC method treats obstacles as agents and makes a zero-sum game with AUV. The obstacle size, shape, number and location are updated by particle swarm optimization (PSO) [46]. As the gaming process proceeds, the level of obstacle placement improves intelligently with the AUV path planning strategy, which enhances the diversity of the simulation environment and improves the training effect.

### 2.1. Preliminaries

The GSAC method consists of the simulation environment, decision agent, environment agent and AUV reward function, as shown in Figure 1. The environment agent is an update strategy for the obstacle parameters (number, size, shape and location), so that the environment evolves gradually from simple to complex with the training process. The decision agent is used to generate a path planning policy applicable to the environment. The reward function aims to help the update of the decision agent and environment agent. It gives a reward to AUV after the performing action. There is a zero-sum game between the environment agent and decision agent. The decision agent aims to increase the action value $Q_{auv}$ obtained in the simulation environment by changing the AUV path planning strategy. The environment agent aims to increase the action value $Q_{env}$ is obtained by changing the parameters of the obstacles in the simulation environment. The AUV reward is the connection of the game relationship between the two sides of the game.

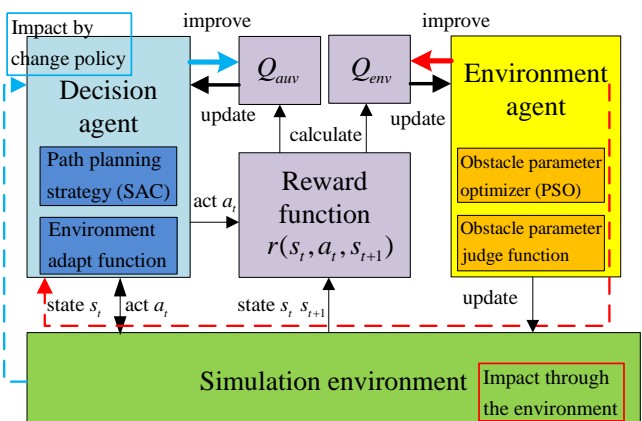

**Figure 1.** GSAC framework.

The decision agent consists of a path planning strategy (using the SAC method) and environment adaptation function. The environment adaptation function is responsible for judging the adaptation of the path planning strategy to the current simulation environment. It can make the update of the decision agent turn off when the adaptation degree reaches the requirement. The environment agent consists of the obstacle parameter optimizer (using the PSO method) and the obstacle parameter judgment function. The obstacle information judgment function is responsible for counting the number of updates of the environment agent and calculating the obstacle size and number. The obstacle information optimizer optimizes the obstacle positions and shapes according to the obstacle sizes and numbers. It can generate the obstacle parameters that make itself obtain the most reward. The decision agent and environment agent are updated alternately during the game.

### 2.2. Underwater Environment Model

The underwater environment includes the obstacles, goal, and starting point, as shown in Figure 2. The coordinate system of the underwater environment employs the geodetic coordinate system. The underwater environment is a two-dimensional space with a square boundary (the side length is 500 m), and the starting point is the coordinate center $O$. The positive direction of $x$-axis is the longitude increment direction, and the $x$-axis passes through the coordinate point $O$. The positive direction of $y$-axis is the latitude increment direction, and the $y$-axis passes through the coordinate point $O$. To prevent the AUV path planning method from being blocked by obstacles and reaching the target, a square centered on the starting point and a square centered on the target point are set as the area without obstacles (the side length is 100m). $\chi_{env} = \{O_n, S_p, E_p, \delta\}$ is defined as the underwater environment. $O_n$ represents the number of obstacles. Considering that the shape of the obstacle is only rectangular, the maximum number of obstacles is set as two to form a more complex shape. $S_p$ represents the starting point position. $E_p$ represents the goal position. $\delta = \bigcup_{i=1}^{O_n} \delta_i$ is defined as obstacles in the environment. $\delta_i = \{e_x, e_y, e_\theta, e_\varphi, e_l\}$ is defined as a single obstacle in the environment. $e_x$ represents the relative distance between the center of the obstacle and the coordinate point $O$ on the $x$-axis. $e_y$ represents the relative distance between the center of the obstacle and the coordinate point $O$ on the $y$-axis. $e_l$ represents the diagonal length of the obstacle. The dashed rectangle in the figure represents the obstacle without the rotation transformation, which becomes the final obstacle after rotating $e_\theta$ around the center point, as shown in the solid rectangle. $e_{th}$ represents the length-width ratio of the rectangular obstacle.

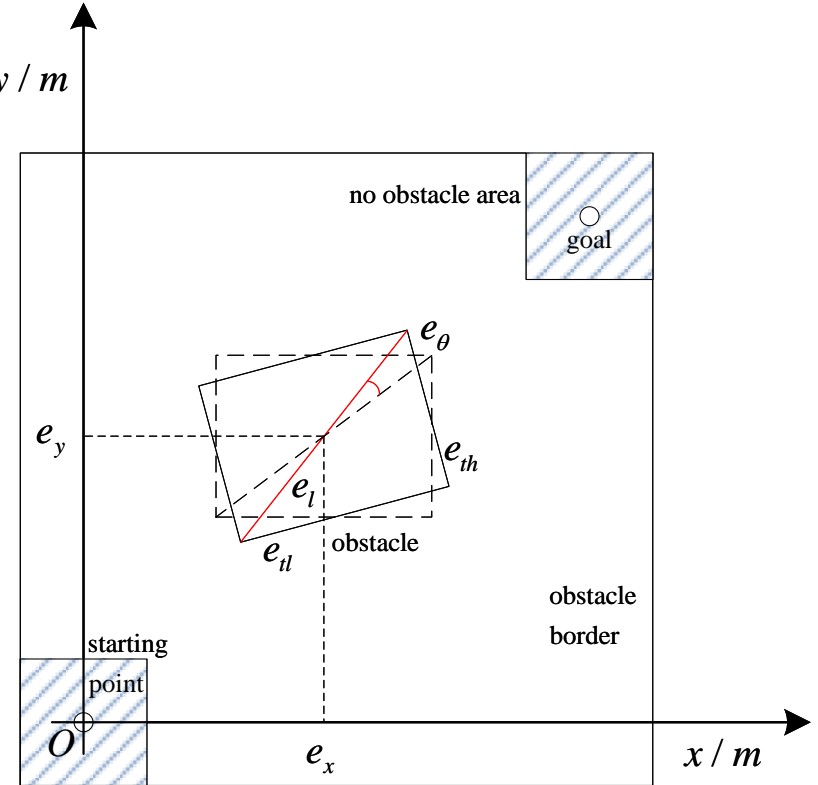

**Figure 2.** Environment schematic.

AUV state is the input of the path planning method, including environment and sensor information. $s_t = \{e_t, o_t\}$ is defined as the AUV state at time t, as shown in Figure 3. $e_t = \{s_t^\theta, s_t^x, s_t^y, s_t^v\}$ is defined as the environment information of AUV. $s_t^\theta$ represents the angle between the AUV direction and the target direction. The target direction refers to the direction in which the AUV center points to the target, and the navigation direction refers

to the current velocity direction of the AUV. $s_t^x$ represents the relative distance between the AUV and coordinate point $O$ on the $x$-axis, $s_t^y$ represents the relative distance between the AUV and coordinate point $O$ on the $y$-axis, and $s_t^v$ represents the velocity of the AUV in the path planning space.

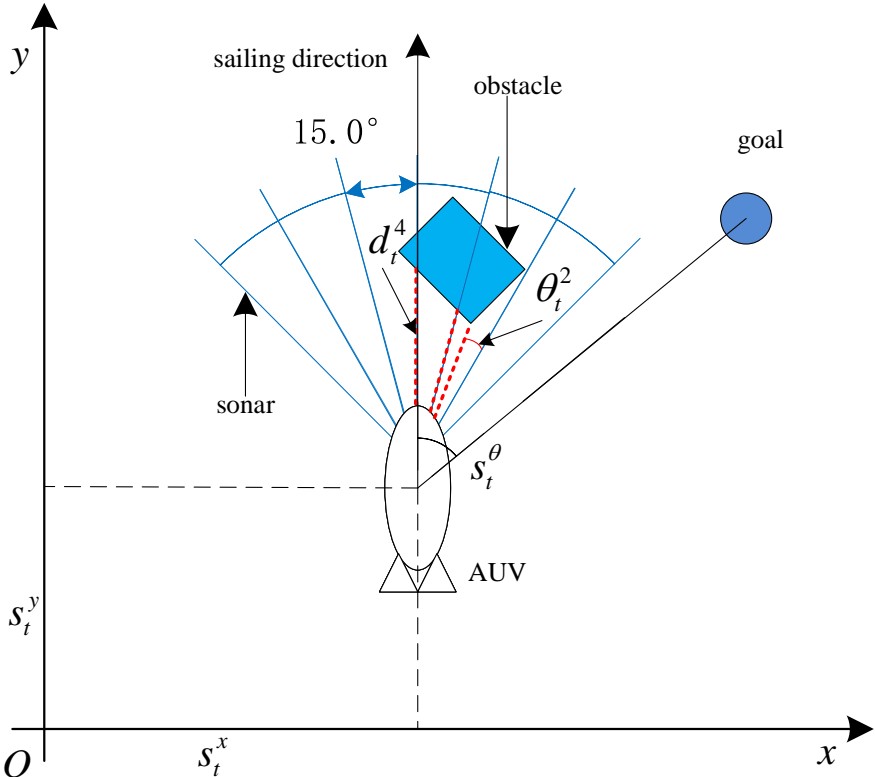

**Figure 3.** Environment schematic.

$o_t$ denotes the observed information of the AUV in the current environment, satisfying the following assumptions:

(1) The environmental information of the AUV is part of the current environment observed by the AUV sensor, and includes all the information required for the AUV path planning decision.

(2) The AUV sensor samples once per sampling cycle (1s) to acquire the current environmental information. The AUV path planning method outputs the action after acquiring the current environment information $e_t$ and observing information $o_t$, and executes it until obtaining the environmental information of the next sampling cycle.

$$o_t = \bigcup_{i=1}^{6} \left( d_t^i, \theta_t^i \right) \tag{1}$$

$o_t$ is the sensor information in the real underwater environment. It is the observed value calculated based on the obstacles in simulated environment, as shown in Equation (1). The AUV sonar is carried in front of the AUV, and the detection area of the sonar is from the northwest to the northeast of the AUV sailing direction. The detection range of the sonar is 50 m. To speed up the training of the AUV path planning method, the sonar detection area is discretized. It is uniformly divided into six blocks, which are named as zones 1 to 6 in a clockwise direction. $\theta_t^i$ represents the angles of the nearest obstacle detected in the zone $i$ at time $t$. $d_t^i$ represents the distances of the nearest obstacle detected in the zone $i$ at time $t$. If no obstacle is detected in the block $i$, the $d_t^i$ and $\theta_t^i$ are constant.

AUV action is the output of the path planning method, including velocity and yaw angle. $a_t = \{v_t, \theta_t\}$ is defined as the AUV action in the current state $s_t$. $v_t$ represents AUV

current velocity in the range $[-1\ \text{kn}, 5\ \text{kn}]$. $\theta_t$ represents AUV current yaw angle in the range $[-15°, 15°]$.

## 3. GSAC Algorithm

### 3.1. Game-Based Optimization Objective

The SAC algorithm object to discover the strategy with the optimal action value $Q(s_t, a_t)$, as shown in Equation (2).

$$J_{SAC} = \arg\max_{\pi} [Q(s_t, a_t) - \alpha_t log\pi(a_t|s_t)] \tag{2}$$

where $\alpha_t$ denotes the adaptive weight called the entropy parameter, $log\pi(a_{t+1}|s_{t+1})$ denotes the action entropy of the AUV at the moment $s_{t+1}$. The expected reward after the agent performs the action is represented by the action value function $Q(s_t, a_t)$, and its update target is shown in Equation (3).

$$\hat{Q}(s_t, a_t) = \mathbb{E}_{p\sim p_\delta(s_{t+1}|s_t, a_t)} r_{auv}(s_t, a_t, s_{t+1}) + \gamma Q(s_{t+1}, a_{t+1}) - \gamma\alpha_t log\pi(a_{t+1}|s_{t+1}) \tag{3}$$

where $\hat{Q}(s_t, a_t)$ denote the update target of $Q(s_t, a_t)$, $r_{auv}(s_t, a_t, s_{t+1})$ denote the reward obtained by the AUV after performing action $a_t$ in the current state $s_t$, $\gamma$ is a constant called the discount rate, $Q(s_{t+1}, a_{t+1})$ is the value of the action value function at the next moment, and $p_\delta(s_{t+1}|s_t, a_t)$ denote the probability of the environment transfers to state $s_{t+1}$ when the obstacle parameter is $\delta$.

The action value $Q(s_t, a_t)$, strategy $\pi(a_t|s_t)$, and action entropy $\alpha_t$ in Equation (3) are updateable parts updated with SAC. $r_{auv}(s_t, a_t, s_{t+1})$ is designed by the designer. Due to the high risk and cost of the AUV field experiments, the AUV path planning needs to be trained in the simulation environment; therefore, obstacle parameters $\delta$ in the simulation environment need to be designed based on the designer's experience. Due to the complexity and diversity of the marine environment and the limitation of the human cognition of the ocean, the manually designed obstacle parameters $\delta$ are difficult to ensure the diversity of the simulation environment and meet the requirements of reinforcement learning methods. To solve this problem, a game-based optimization objective is proposed, as shown in Equations (4) and (5).

$$J_{env} = \arg\max_{\delta} \mathbb{E}_{p\sim p_\delta(s_{t+1}|s_t, a_t), t\in\tau, a_t\sim\arg\max_{\pi} J_{auv}} \left[\overline{Q_{env}(s_t, a_t)}\right] \tag{4}$$

$$J_{auv} = \arg\max_{\pi} \mathbb{E}_{p\sim p_{\delta env}(s_{t+1}|s_t, a_t)} [Q_{auv}(s_t, a_t) - \alpha_t log\pi(a_t|s_t)] \tag{5}$$

where $\delta$ env denotes the obstacle parameters generated by environment agent, $J_{env}$ denotes the decision agent target, $\tau$ denotes the AUV trajectory, $J_{auv}$ denotes the environment agent target, $\overline{Q_{env}}$ denotes the average of $Q_{env}$. $Q_{env}(s_t, a_t)$ denote the action value function of AUV, as shown in Equation (6). $Q_{auv}(s_t, a_t)$ denote the action value function of environment agent, as shown in Equation (7).

$$Q_{env}(s_t, a_t) = \mathbb{E}_{p\sim p_{\delta env}(s_{t+1}|s_t, a_t), t\in\tau, a_t\sim\pi} \sum_{t=1}^{n} \gamma^{t-1} r_{env}(s_t, a_t, s_{t+1}) - \alpha_t log\pi \tag{6}$$

$$Q_{auv}(s_t, a_t) = \mathbb{E}_{p\sim p_{\delta env}(s_{t+1}|s_t, a_t), t\in\tau, a_t\sim\pi} \sum_{t=1}^{n} \gamma^{t-1} r_{auv}(s_t, a_t, s_{t+1}) - \alpha_t log\pi \tag{7}$$

where $r_{env}(s_t, a_t, s_{t+1})$ denotes environment agent reward, $r_{auv}(s_t, a_t, s_{t+1})$ denotes AUV reward.

In the game-based optimization objective, the action value $Q(s_t, a_t)$, strategy $\pi(a_t|s_t)$, action entropy $\alpha_t$, and simulation environment are the updateable parts updated with GSAC. In the update objective, GSAC considers both sides of the game, so that the environment agent is adaptively updated based on the AUV path planning method. The improved

environment agent creates more intelligent obstacle parameters to increase the simulation environment's diversity and enhance the training effect of the AUV path planning technique. The environment reward $r_{env}(s_t, a_t, s_{t+1})$ and the AUV reward $r_{auv}(s_t, a_t, s_{t+1})$ need to be designed. The design of the AUV reward is the same as that of SAC. The sum of the environment reward and AUV reward is zero due to the zero-sum game between the environment agent and decision agent. GSAC is proposed to increase the diversity of the simulation environment through a zero-sum game between the environment agent and the decision agent, thus enhancing the simulation environment's training effect. As shown in Equation (4), the obstacle parameters are updated according to the AUV strategy $\pi(a_t|s_t)$ and the environment action value $Q_{env}(s_t, a_t)$. When the AUV strategy is weak at the start of training, the obstacle arrangement is simple. The obstacle arrangement becomes more complex as the enhance of AUV strategy. In the process of solving the zero-sum game, the AUV is trained in a simulation environment that changes with its own strategy. The proposed method enhances the adaptability and convergence speed of the AUV in unknown environments.

### 3.2. Environment Agent

The environment agent optimizes the obstacle parameters according to Equation (4). The optimization objective $J_G$ requires the total environment reward of the AUV path planning trajectory in the simulation environment with obstacle parameters' $\delta$. The computational cost of each update of the obstacle parameters is the same as that of the AUV to complete a path planning task. Since RL requires a lot of trial and error, it is computationally too expensive for the environmental agent to use the RL-based approach. Since the particle swarm algorithm (PSO) requires low computational cost and high optimization capability for this problem, it is used as the optimization method for the environment agent.

The obstacle parameters in the simulation environment are divided into obstacle size, shape, number and location. The values of obstacle size and number are calculated by the obstacle parameter judgment function. The calculation of the obstacle parameter judgment function is shown in Equation (8).

$$\begin{cases} e_l = d_l * \left\lfloor \frac{n}{n_l} \right\rfloor \\ n_o = \left\lfloor \frac{n}{n_n} \right\rfloor \end{cases} \tag{8}$$

where n is the number of updates of the environment agent, $d_l$ is the obstacle size update weight, $n_l$ is the obstacle size update threshold, $n_n$ is the obstacle number update threshold, $\lfloor \rfloor$ is the downward rounding function.

The PSO algorithm is used to optimize the obstacle parameter $\delta$ the during environment agent update. It inputs the size and number of obstacles obtained according to the environmental agent update process, and outputs the optimal obstacle parameter $\delta_{min}$. The environment agent starts the update when the decision agent ends the update. After the $n_p$ times update, environment agent output $\delta_{min}$.

The environment agent initializes the first generation of obstacle parameters $x^1$ and velocity $v^1$. $n_x$ sets of the initialized obstacle parameters are included in the obstacle parameters' $x^1$. Velocity $v^1$ is an adaptive parameter responsible for $x^1$ update. The environment agent calculates the target value $J_G$ according to each group of obstacle parameters in $x^1$, respectively, and updates $x_{pb}$ and $x_{gb}$ according to the target value $J_G$. The dimension of $x_{pb}$ is the same as $x^1$, which is the statistics of the individual with the optimal target value in each group of obstacle parameters. $x_{gb}$ is the obstacle parameter with the optimal target value in this update of the environment agent. The environment agent updates the obstacle parameters $x^{t+1}$ and velocity $v^{t+1}$ based on $x^t$ and $v^t$, which are calculated as shown in Equation (9).

$$\begin{cases} v^{t+1} = v^t + c_1 r_1 \left( x_{pb} - x^t \right) + c_2 r_2 \left( x_{gb} - x^t \right) \\ x^{t+1} = x^t + v^{t+1} \end{cases} \tag{9}$$

where $c_1$ and $c_2$ are the update weights, and $r_1$ and $r_2$ are the random numbers.

The update pseudo code of the environment agent is shown in Algorithm 1.

---

**Algorithm 1** Environment Agent Update

---

**Input:** Environment agent update times $n$

1: Compute:$e_l = d_l * \left\lfloor \frac{n}{n_l} \right\rfloor , n_o = \left\lfloor \frac{n}{n_n} \right\rfloor$

2: Initialize $x^1$ according to $e_l$, $n_o$
   Initialize $v^1, x_{pb}, x_{gb}, J_G(x^0), J_G(g), i = 1, j = 1$

3: **while** $j < n_p$ **do**

4:     **for** each obstacle parameter in $x^i$, compute $J_G(x_n^i)$ **do**

5:         **if** $J_G(x_n^{i-1}) > J_G(x_n^i)$ **then**

6:             $x_{pb}^i \leftarrow x_n^i, J_G(x_n^{i-1}) \leftarrow J_G(x_n^i)$

7:         **end if**

8:         **if** $J_G(g) > J_G(x_n^i)$ **then**

9:             $x_{gb} \leftarrow x_n^i, J_G(g) \leftarrow J_G(x_n^i)$

10:        **end if**

11:     **end for**

12:     Randomly initialize $r_1, r_2$

13:     Compute: $v^{i+1} = v^i + c_1 r_1 \left( x_{pb} - x^i \right) + c_2 r_2 \left( x_{gb} - x^i \right), x^{i+1} = x^i + v^{i+1}$

14:     $i \leftarrow i + 1, j \leftarrow j + 1$

15: **end while**

**Output:** $\delta_{min} \leftarrow x_{gb}$

---

*3.3. Reward Function*

Reward function is crucial in RL. The AUV path planning reward is typically the sparse reward because it mainly focused on the end of mission. Sparse rewards have a bad effect to the SAC train. To solve the sparse reward problem in AUV path planning, a continuous modular reward function is designed in the GSAC. The reward function takes into account several aspects such as AUV obstacle avoidance, arrival at the target, and stability of AUV control. The reward function is calculated as shown in Equation (10).

$$\begin{cases} r_{auv}(s_t, a_t, s_{t+1}) = \tau_1 r_1 + \tau_2 r_2 + \tau_3 r_3 + \tau_4 r_4 + \tau_5 r_5 \\ r_{env}(s_t, a_t, s_{t+1}) = -(\tau_1 r_1 + \tau_2 r_2 + \tau_3 r_3 + \tau_4 r_4 + \tau_5 r_5) \end{cases} \tag{10}$$

where $r_{auv}(s_t, a_t, s_{t+1})$ denotes AUV reward, $r_{env}(s_t, a_t, s_{t+1})$ denotes environment reward, $\tau_1, \tau_2, \tau_3, \tau_4,$ and $\tau_5$ denote the weight of each reward module, $r_1, r_2, r_3, r_4,$ and $r_5$ denote the reward module.

The reward of AUV safety is mainly reflected in the safety reward module $r_1$. After the AUV performs the action, the reward is calculated based on the distance to the nearest obstacle detected by sonar. If the nearest obstacle distance is less than the safety threshold $d_1$ but greater than the warning threshold $d_2$, the fixed penalty is obtained; if the nearest obstacle distance is less than the warning threshold $d_2$, the fixed penalty is obtained. The calculation of the target reward module $r_1$ is shown in Equation (11).

$$r_1 = \begin{cases} 0, & if\ d_s > d_1 \\ -0.5, & if\ d_2 > d_s \geq d_1 \\ -1, & if\ d_2 \geq d_s \end{cases} \tag{11}$$

where $d_s$ is the distance between AUV and the nearest obstacle detected by perception module at state $s_{t+1}$, $d_1$ is a constant called the safety threshold, and $d_2$ is a constant called the warning threshold.

The reward of AUV speed is mainly reflected in the speed reward module of $r_2$. After the AUV performs the action, if the AUV is closer to the target point, it will get a reward

proportional to the current speed. The calculation of the speed reward module $r_2$ is shown in Equation (12).

$$r_2 = \begin{cases} \frac{v_{now}}{5}, & if\ d_t > d_{t+1} \\ 0, & if\ d_t \le d_{t+1} \end{cases} \tag{12}$$

where $v_{now}$ is the current speed of AUV, $d_t$ is the distance between AUV and target point at state $s_t$, and $d_{t+1}$ is the distance between the AUV and target point at state $s_{t+1}$.

The reward of the AUV target is mainly reflected in the target reward module $r_3$. After the AUV performs the action, the AUV will obtain a fixed reward and additional rewards if it is closer to the target, otherwise it will obtain a fixed punishment. Additional rewards are calculated according to the contribution of the current AUV action to the target. The smaller the difference between the course of the AUV and the direction of the target point after the action is executed, the greater the contribution of the current action to the target, so the more additional rewards the AUV will receive. The calculation of the target reward module $r_3$ is shown in Equation (13).

$$r_3 = \begin{cases} 4 - \frac{\Delta\theta}{\pi} \times 6, & if\ d_t > d_{t+1} \\ -1, & if\ d_t \le d_{t+1} \end{cases} \tag{13}$$

where $\Delta\theta$ is the difference between the course of the AUV and the direction of the target point.

The reward of the AUV mission completion is mainly reflected in the mission completion module $r_4$. After the AUV performs the action, if the distance from the target point is less than the target point range, it reaches the target point area and obtains a fixed reward; if the nearest obstacle detected by sonar is less than the dangerous distance, it will be judged as colliding with the obstacle, the task failure, and s fixed value will be given. The calculation of the target reward module $r_4$ is shown in Equation (14).

$$r_4 = \begin{cases} 100, & if\ reach\ goal \\ -100, & if\ touch\ collision \end{cases} \tag{14}$$

The reward of AUV stability is mainly reflected in the stability module $r_5$. In order to make the AUV navigation process more stable, this study set up a stability reward. The more the AUV's action $a_t$ affects its speed and direction, the smaller the stability reward. The calculation of the stability reward module $r_5$ is shown in Equation (15).

$$r_5 = -2 \times \left( (\Delta\theta)^2 + (\frac{\Delta v}{6})^2 \right) \tag{15}$$

where $\Delta\theta$ is the change of AUV direction after perform action $a_t$, and $\Delta v$ is the change of AUV speed after perform action $a_t$.

### 3.4. Decision Agent

The schematic of the decision agent is shown in Figure 4. The decision agent is divided into five components, namely Actor, Critic, Entropy term, Replay buffer and environment adaptation function. Actor is responsible for making path planning decisions, generating the policy $\pi\left(a_t \middle| s_t^k, \theta^\pi\right)$ based on the current environment state $s_t$, and sampling the action $a_t$ according to the policy. Critic is responsible for evaluating the actions generated by Actor and assisting Actor in updating. The purpose of the Entropy term is to calculate the action entropy $\alpha_t log\pi\left(a_t \middle| s_t^k, \theta^\pi\right)$ of the current moment action and assist the update of Actor and Critic. The components of the AUV path planning agent can be divided into two modules, namely decision module and training module. The decision-making module is the most critical module and is mainly responsible for AUV decision making. The decision-making module performs the functions of Actor, as shown in the red solid box in Figure. The training module is mainly used to assist the update of decision-making module, as shown in the blue solid box in the figure.

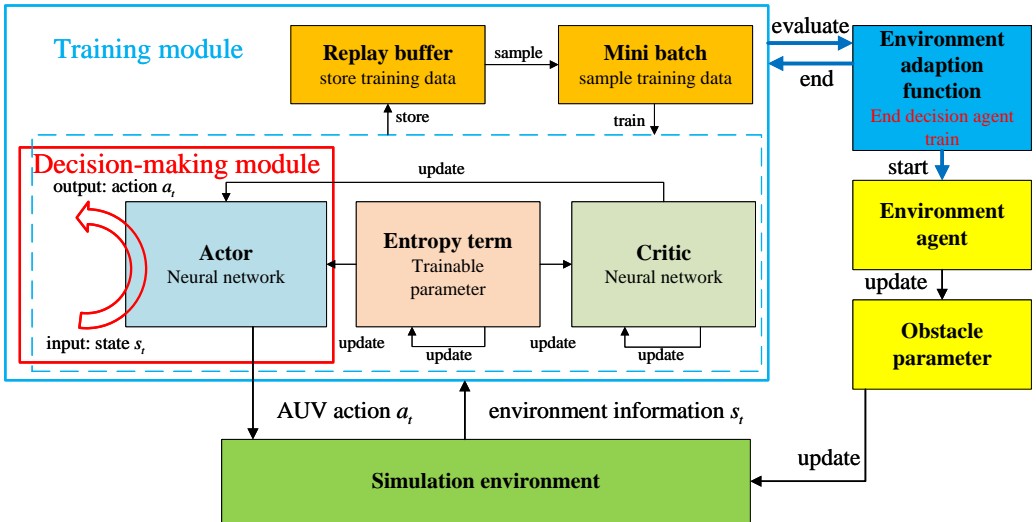

**Figure 4.** Decision agent schematic.

The environmental fitness function can control the training of the decision agent to turn off and the update of the environmental agent to turn on. It is shown in Equation (16). When the output value of the environment adaptation function is 1, the training of the AUV path planning agent is turned off, and the update of the environment agent is turned on. When the output value of the environment fitness function is 0, the AUV path planning agent continues training.

$$f_s = \begin{cases} 1, & if \ e_s \geq e_{m1} \\ 1, & if \ e_e \geq e_{m2} \\ 0, & else \end{cases} \tag{16}$$

where $e_s$ is the number of times the AUV has reached the target in the past 100 episodes, $e_e$ is the number of times the AUV has been trained in the current environment, and $e_{m1}$ and $e_{m2}$ are constants.

The update details of the AUV path planning agent are shown in Figure 5. The AUV path planning agent is mainly divided into a decision-making module and training module. The decision-making module, as shown by the black arrow in Figure, mainly includes the decision making and the acquisition of the training samples when completing the task. The training module mainly uses the samples obtained from the decision-making module for training, as shown by the red arrow in figure. The Replay buffer is an important connection between the decision-making module and the training module. It is responsible for storing the training samples collected by decision-making module, and sampling $m$ data during the training process to train the RL method. Replay buffer sets the maximum amount $\varnothing$ of stored data. When the maximum data storage capacity is exceeded, the new data will replace the original data.

Actor adopts a policy network with parameters $\theta^\pi$ to represent the stochastic policy of the AUV. The policy network is the core of the AUV path planning method. As a neural network, it inputs the current environment information of the AUV, and outputs the Gaussian distribution of action strategies. When making path planning decisions, the policy network outputs the current strategy based on the current environment information, and the AUV obtains the current action based on the current strategy distribution. The goal of the policy network update is to minimize the objective $J_\pi(\theta^\pi)$. The calculation of $J_\pi(\theta^\pi)$ is shown in Equation (17).

$$J_\pi(\theta^\pi) = \mathbb{E}_{a_t \sim \pi, p \sim p_{\delta env}(s_{t+1}|s_t, a_t)} \left[ Q\left(s_t^k, a_t^{new}\right) - \alpha_t log \pi \left(a_t^{new} \middle| s_t^k, \theta^\pi\right) \right] \tag{17}$$

where $\theta^\pi$ is a policy network parameter, and $a_t^{new}$ is the newly generated action of the policy network according to the current state.

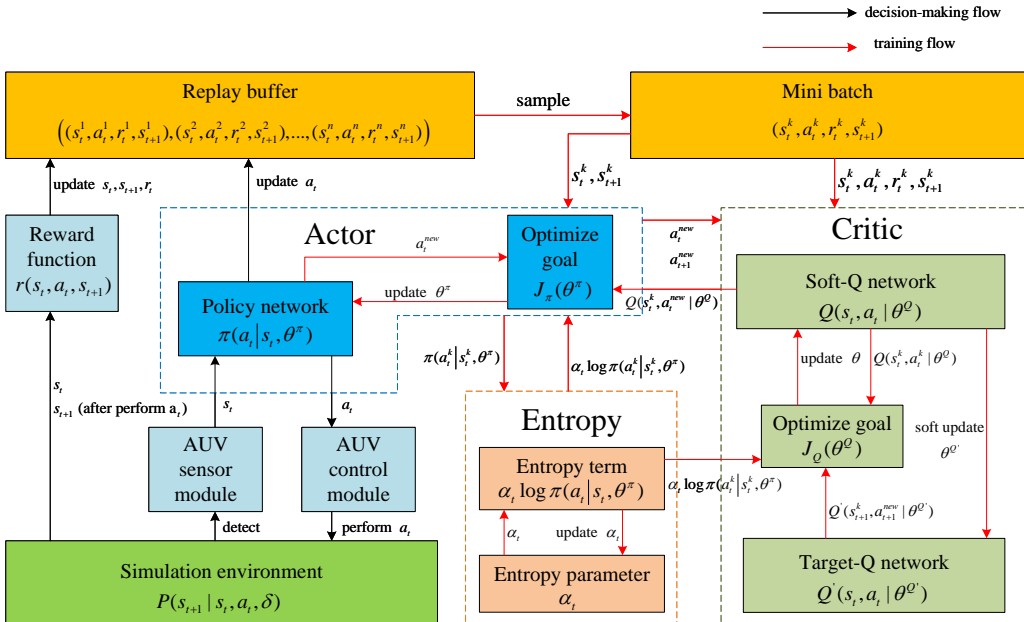

**Figure 5.** Decision agent update schematic.

Critic includes two neural networks, the Soft-Q network and Target-Q network. They have the same initial parameters and network structure. The input is the current environment information of the AUV and the current action of the AUV, and their outputs are the action value functions. The Soft-Q network is used to update the policy network. The Target-Q network aims to reduce the effect of the estimation bias on the update of the Soft-Q network.

The Soft-Q network with parameters estimates the action value function with entropy terms based on the current state and action. The goal of the soft-q network update is to minimize the target $J_Q\left(\theta^Q\right)$. The calculation of $J_Q\left(\theta^Q\right)$ is shown in Equation (18).

$$J_Q\left(\theta^Q\right) = \mathbb{E}_{p \sim p_{\delta env}(s_{t+1}|s_t, a_t)}\left[\left(Q(s_t^k, a_t^k|\theta^Q) - \hat{Q}\right)^2\right] \tag{18}$$

where $\delta env$ is the obstacle parameter generated by the environment agent, $\theta^Q$ is the Soft-Q network parameter, and $\hat{Q}$ is the update target. The calculation of $\hat{Q}$ is shown in Equation (19).

$$\hat{Q} = r_t^k + \gamma\mathbb{E}_{p \sim p_{\delta env}(s_{t+1}|s_t, a_t)}\left[Q\prime\left(s_{t+1}^k, a_{t+1}^{new}\Big|\theta^{Q\prime}\right) - \alpha_t log \pi\left(a_{t+1}^{new}\Big|s_{t+1}^k, \theta^\pi\right)\right] \tag{19}$$

where $r_t^k$ is AUV reward at time $t$, $Q\prime\left(s_{t+1}^k, a_{t+1}^{new}|\theta^{Q\prime}\right)$ is an estimate of the Target-Q network, $\theta^{Q\prime}$ is the Target-Q network parameter, $\pi\left(a_{t+1}^{new}\Big|s_{t+1}^k, \theta^\pi\right)$ is a new policy generated by the policy network based on the state, and $a_{t+1}^{new}$ is a current action obtained by sampling from the new policy.

Target-Q network with parameters estimates the target of Soft-Q network based on states and actions. The target-Q network has the same initial parameters as the Soft-Q network, but the update method is different. It performs a soft update with the Soft-Q network parameters as the target. The soft update of the Target-Q network is shown in Equation (20).

$$\theta^{Q\prime} = \tau_s\theta^Q + (1 - \tau_s)\theta^{Q\prime} \tag{20}$$

where $\tau_s$ is a weight called the soft update rate.

To improve the exploration ability of path planning strategies, action entropy and entropy parameters are introduced. Action entropy is the information entropy of the AUV strategy, which represents the randomness of the AUV strategy. Entropy parameter is a coefficient of action entropy. As an adaptive parameter, it indicates the importance of the action entropy in the update of Actor and Critic updates. Entropy parameter represents the randomness of the strategy and is updatable. The larger the entropy parameter is, the greater is the randomness of the actions generated by the AUV path planning method. The goal of the entropy parameter update is to minimize the objective $J_\alpha$. $J_\alpha$ is calculated as shown in Equation (21).

$$J_\alpha = -\alpha_t log\pi\left(a_t^{new}\middle|s_t^k, \theta^\pi\right) - \alpha_t H \tag{21}$$

where $H$ is a constant.

GSAC method (Algorithm 2) is obtained by combining the SAC method with Algorithm 1.

---

**Algorithm 2** GSAC Algorithm

---

1: Randomly initialize Policy network $\pi(a_t|s_t, \theta^\pi)$, Soft-Q network $Q(s_t, a_t|\theta^Q)$
2: Initialize entropy parameter $\alpha_t = -\log(2)$, $f_s = 0$
　 Initialize Target-Q network $Q\prime(s_t, a_t|\theta^{Q\prime})$, initialize $\theta^{Q\prime} = \theta^Q$
3: **while** not converge **do**
4: 　**while** $f_s = 0$ **do**
5: 　　Initialize simulation environment for exploration
6: 　　**while** not reach goal **do**
7: 　　　Receive initial observation state $s_t$
8: 　　　Select action $a_t$ $\pi(a_t|s_t, \theta^\pi)$ according to current policy
9: 　　　Take action $a_t$ and observe state $s_{t+1}$
10: 　　　Compute $r_{auv}(s_t, a_t, s_{t+1})$ according to Equation (10)
11: 　　　Store $(s_t, a_t, r_{auv}, s_{t+1})$ into Replay buffer
12: 　　　Update Policy network, Soft-Q network, Entropy parameter, Target-Q network according to Equations (17)–(21)
13: 　　**end while**
14: 　Optimize $\delta_{min}$ according to Algorithm 1
15: 　Update simulation environment according to $\delta_{min}$
16: 　Compute $f_s$ according to Equation (16)
17: 　**end while**
18: **end while**
**Output:** Policy network, Soft-Q network, Entropy parameter, Target-Q network

---

## 4. Simulation Results

To verify the feasibility of the GSAC method, a simulation training environment and a test environment are set up with the Python programming language. Meanwhile, different training environments are set up for GSAC and comparison algorithms based on different training methods. Comparison algorithms include SAC and DQN [26]. DQN's policy is the $\varepsilon$-greedy policy. After training in the training environment, put GSAC, SAC and DQN into the same test environment for testing.

### 4.1. Train Environment

In this study, a simulation training environment is constructed for GSAC. The AUV is modeled as a mass with controllable velocity in the simulation environment. The size of the simulation training environment is $500 \times 500$ m, the initial position of the AUV is located at the coordinate point (50 m,50 m) in the figure, and the target center is located at the point (450 m, 450 m). The environment is clear with a boundary. When the GSAC method is trained, the environment agent adds obstacles to the environment or transforms the size, shape and position of obstacles. The obstacles in the environment are rectangular.

The hyperparameters of GSAC, SAC and DQN are determined based on references [27,29,40,46,47] and engineering experience, as shown in Table 1.

**Table 1.** Hyperparameters of GSAC, SAC and DQN.

| Hyperparameter | Value |
| --- | --- |
| Learning rate $\alpha_1$, $\alpha_2$ | 0.0001 |
| Discount factor $\gamma$ | 0.99 |
| Replay buffer size $\varnothing$ | 102400 |
| Mini batch size $m$ | 128 |
| Soft update frequency $\tau_s$ | 0.01 |
| Max episode num $M$ | 6000 |
| Reward weight $\tau_1, \tau_2, \tau_3, \tau_4, \tau_5$ | 1 |
| PSO weight $c_1$ | 0.49445 |
| PSO weight $c_2$ | 1.49445 |
| Population size $n_x$ | 150 |
| Explore rate $\varepsilon$ | 0.9 |

The parameters of the environment agent in GSAC are determined based on engineering experience, as shown in Table 2.

**Table 2.** Parameter of environment agent.

| Parameter | Value |
| --- | --- |
| Game success episode $e_{m1}$ | 15 |
| Game max episode $e_{m2}$ | 45 |
| Size update threshold $n_l$ | 12 |
| Amount update threshold $n_n$ | 240 |
| Size update weight $d_l$ | 20 |

Figure 6 demonstrates the training process of GSAC. At episode 1, the AUV fails to reach the target in the open environment. With the training of the decision agent, the AUV can reach the target at episode 31, while the environment agent adds obstacles to the environment. With the training process, the environment agent adds small obstacles to the environment at episode 294, which result in the AUV being unable to reach the target under the interference of the obstacles. The AUV path planning agent updates and improves its strategy, and is rid of the interference of obstacles. After that, the environment agent keeps upgrading the obstacle size along with the game between it and the path planning agent. When the obstacle size exceeds the threshold, it indicates that it is difficult for a single obstacle to prevent the AUV from reaching the target, and the number of obstacles will be increased to 2. At episode 3726, the AUV cannot reach the target due to the interference of two obstacles. With the updating of AUV path planning agent and the improvement of strategy, the AUV can reach the target under the interference of two obstacles at episode 3742.

### 4.2. Simulation Test Results

To verify the path planning capability of AUV path planning strategy in unknown underwater environment, two 2D underwater simulation environments were constructed in a high performance computer using Python programming language, as shown in Figure 7. The continuous unknown underwater test environments were designed by referring to the common scenarios such as canyons, large rocks and reef clusters, mainly including the static test environment and dynamic test environment.

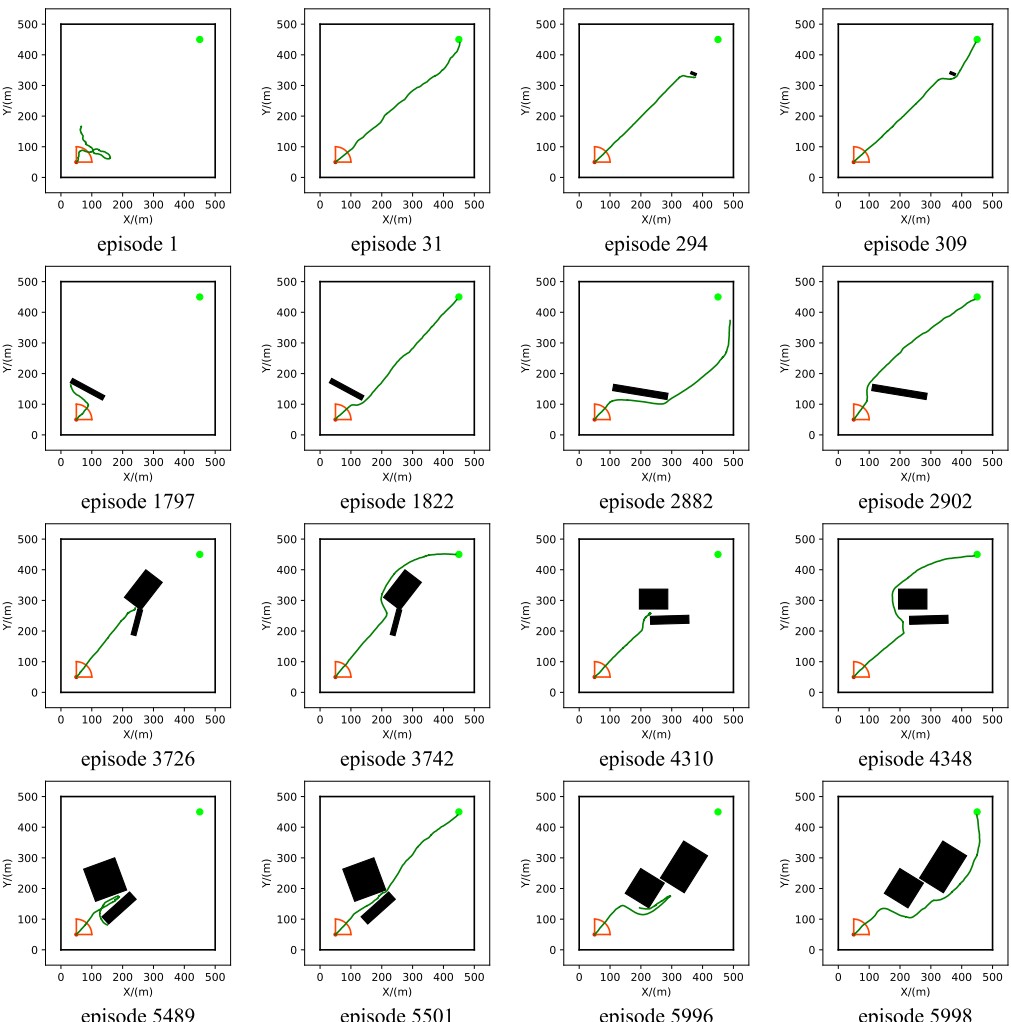

**Figure 6.** Training process of GSAC. ●: AUV; ●: Goal; ■: Obstacle; —: Border; —: Sonar range; —: AUV trajectory.

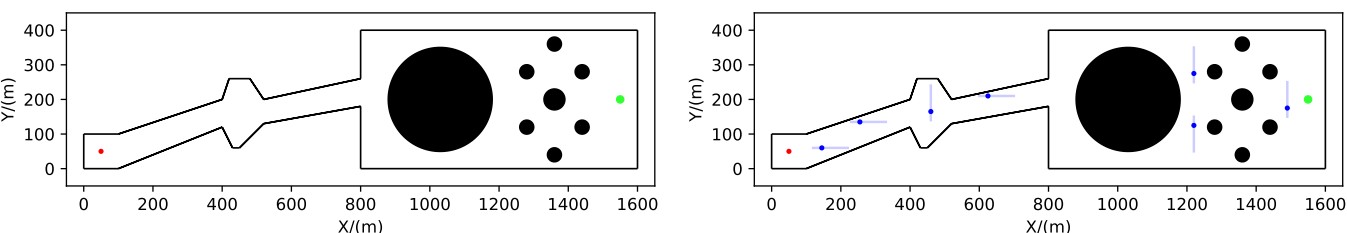

**Figure 7.** Test environment. (**a**) Static environment. (**b**) Dynamic environment. ●: AUV; ●: Goal; ■: Obstacle; ●: Dynamic obstacle; —: Dynamic obstacle trajectory.

The static environment is used to test the path planning performance of the AUV in the unknown static underwater environment. It mainly includes a narrow canyon area located on the left, a large underwater rocky area in the middle, and a submerged reef area on the right. The black lines in the environment are obstacles, the red dots are AUVs, and the green dots are targets. The dynamic environment is a dynamic test environment, which adds small moving obstacles represented by blue circles on the basis of the static environment. It is used to test the path planning performance of the AUV in an unknown dynamic underwater environment. The blue solid line in the environment is the trajectory of the moving obstacle, and the dynamic obstacles move reciprocally in the trajectory. The size of the test environment is 400 × 1600 m, the initial position of the AUV is located at

the point (50 m, 50 m), and the target center point is located at the point (1550 m, 200 m). The parameters of moving obstacles are shown in Table 3.

**Table 3.** Parameters of dynamic obstacles.

| Obstacle | End Point 1 (m) Starting Point | End Point 2 (m) | Speed (m/s) |
|---|---|---|---|
| 1 | (120, 60) | (220, 60) | |
| 2 | (460, 140) | (460, 240) | |
| 3 | (600, 210) | (700, 210) | |
| 4 | (1220, 250) | (1220, 350) | 0.5 |
| 5 | (1220, 150) | (1550, 50) | |
| 6 | (230, 135) | (330, 135) | |
| 7 | (1490, 150) | (1490, 250) | |

After constructing the test environment, this paper tests GSAC, SAC and DQN, and compares the advantages and disadvantages of these methods.The SAC and the DQN are tested after the pre-training, as shown in Figure 8, and the training parameters of them are the same as those in Table 1. SAC and DQN cannot guide AUV to reach the target at episode 1. With continuous training, SAC and DQN can successfully guide AUV to reach the target.

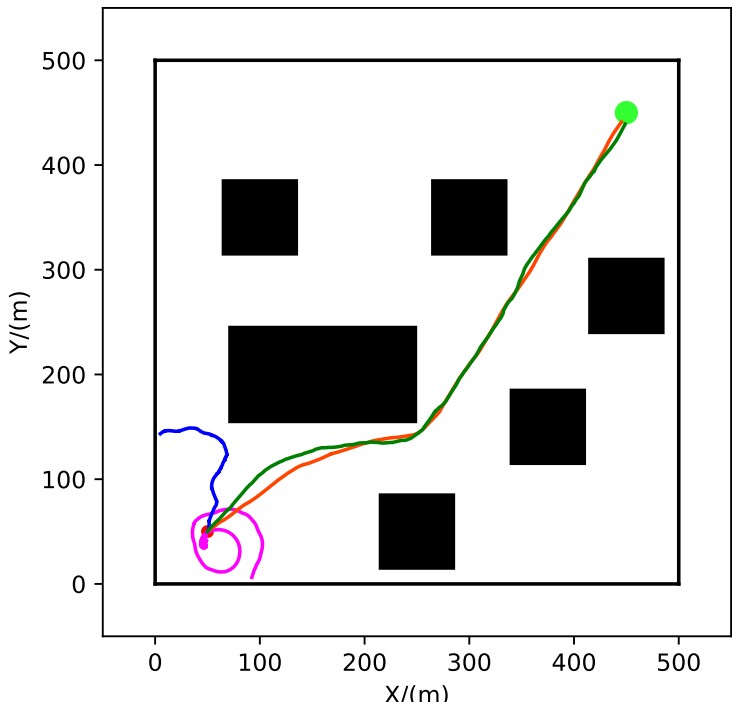

**Figure 8.** Pretraining process of SAC and DQN. •: AUV; •: Goal; ■: Obstacle; —: SAC training episode 1 trajectory; —: SAC training episode 6000 trajectory; —: DQN training episode 1 trajectory; —: DQN training episode 6000 trajectory.

To verify the adaptability of GSAC, SAC and DQN to unknown underwater environments, they are tested in static and dynamic environments, and their test performance is counted. According to the principle of variable control, the hyperparameters of these methods during the test are the same as Table 1. The test performance is divided into two main parameters and three secondary parameters. The main parameters included Success Episodes and Training Episodes. Success Episodes refers to the number of AUV tasks successfully executed in the test. The larger this number, the better the stability of the AUV

path planning strategy. Training Episodes refer to the current number of training rounds. The smaller the number, the faster the training speed of the AUV path planning strategy. The secondary parameters include the Truly Shortest Path Length, Average Length and Yaw Angle. The Truly Shortest Path Length refers to the length of the shortest path found by the AUV during the test. The smaller the number, the better the AUV path planning strategy in finding the optimal path. The Average Length refers to the average path length of the AUV successfully reaching the target during the test. The smaller the number, the more stable the AUV path planning strategy in finding the optimal path. Yaw Angle refers to the average yaw angle of each step in the path of the AUV successfully reaching the target during the test. The smaller the number, the smoother the AUV path planning strategy and the lower the control difficulty.

### 4.2.1. Static Environment Simulation

The test data of GSAC, SAC and DQN in the static environment are shown in Table 4.

**Table 4.** Static environment simulation result.

| Algorithm | Training Episode | Truly Shortest Path Length (m) | Average Length (m) | Yaw Angle (°) | Success Episodes |
|---|---|---|---|---|---|
| GSAC | 1–500 | 1636.61 | 1680.52 | 4.97 | 496 |
| GSAC | 501–1000 | 1630.19 | 1643.17 | 4.92 | 500 |
| SAC | 1–500 | 1620.26 | 1672.65 | 4.96 | 417 |
| SAC | 501–1000 | 1631.23 | 1671.48 | 4.96 | 499 |
| DQN | 1–500 | 1633.29 | 1718.19 | 4.20 | 446 |
| DQN | 501–1000 | 1623.93 | 1683.59 | 4.36 | 387 |

At the same training episodes, the GSAC performs better than SAC and DQN in the Success Episodes. The Truly Shortest Path Length of GSAC at 501–1000 episodes is 1630.19 m, which is 1.04m shorter than that of SAC and 6.26 m and longer than that of DQN. At 501–1000 episodes, the average length of GSAC is 1643.17 m, which is 28.31 m shorter than that of SAC and 40.42 m shorter than that of DQN. At 501–1000 episodes, the Yaw Angle of GSAC is 4.92°, which is 0.04° less than that of SAC and 0.56° more than that of DQN. Due to the influence of $\alpha_t log\pi$ in Equation (17), GSAC and SAC have larger action randomness, so they have a larger yaw angle than DQN.

The curves of GSAC, SAC and DQN in the static test environment are shown in Figure 9. Figure 9a shows the task completion rate curves for the last 100 episodes. After a small amount of trial and error in the initial exploration, GSAC can quickly adapt to the environment and reach the goal. The SAC need 400 episodes to adapt to the static environment. DQN can quickly adapt to the static environment before 400 episodes, but it does not converge at 400-1000 episodes. Figure 9b shows the average reward per step curves. Compared with SAC, GSAC earns higher rewards and has a smoother reward curve. After convergence, SAC's reward curve still exhibits fluctuations. DQN does not converge at 400–1000 episodes, its reward curve still exhibits fluctuations. Figure 9c shows the average yaw angle curves per step. The yaw angle of GSAC is more stable at the convergence. Figure 9d shows the path length curves. A path length of 0 indicates that the AUV did not successfully reach the target in this episode. GSAC has faster convergence, stable task performance, and shorter planned paths. After converge, the SAC's path length curve still exhibits small fluctuations. The DQN's path length curve exhibits fluctuations at 400–1000 episodes. The optimal paths of GSAC, SAC and DQN in the static test environment are shown in Figure 10. Compared with the SAC algorithm, the GSAC can obtain more rewards in the static underwater environment with faster convergence, stability, and robustness. Compared with the DQN algorithm, the GSAC has faster convergence, stability, and robustness in the static underwater environment.

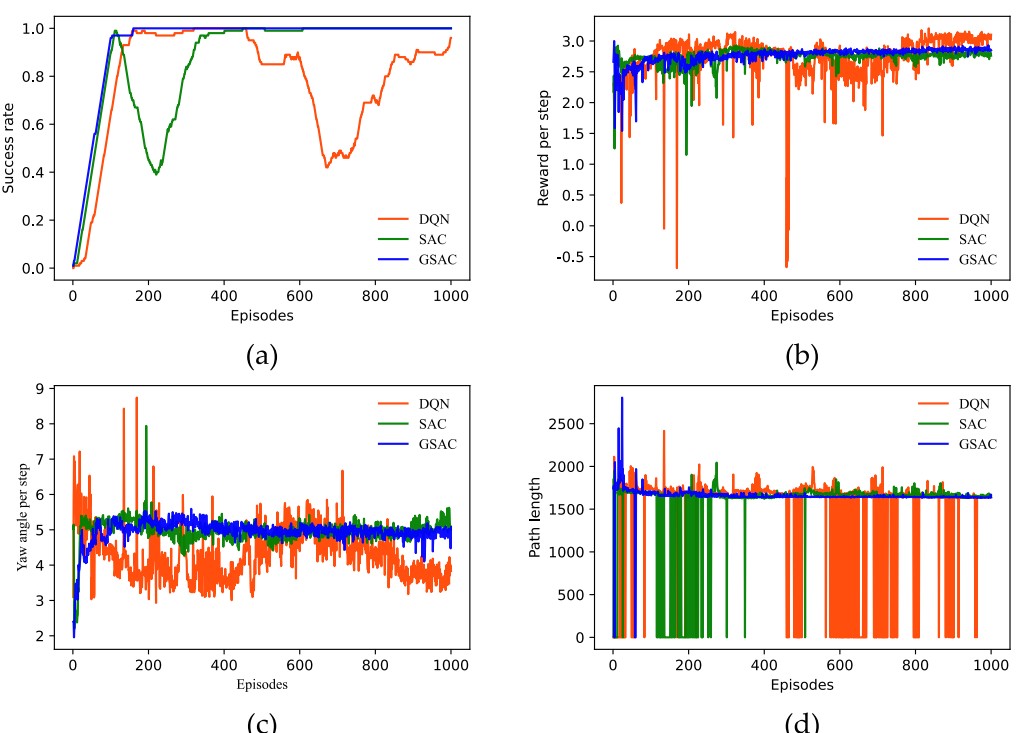

**Figure 9.** Static environment simulation curves. (**a**) Success rate (last 100 episodes).(**b**) Reward per step. (**c**) Yaw angle. (**d**) Path length.

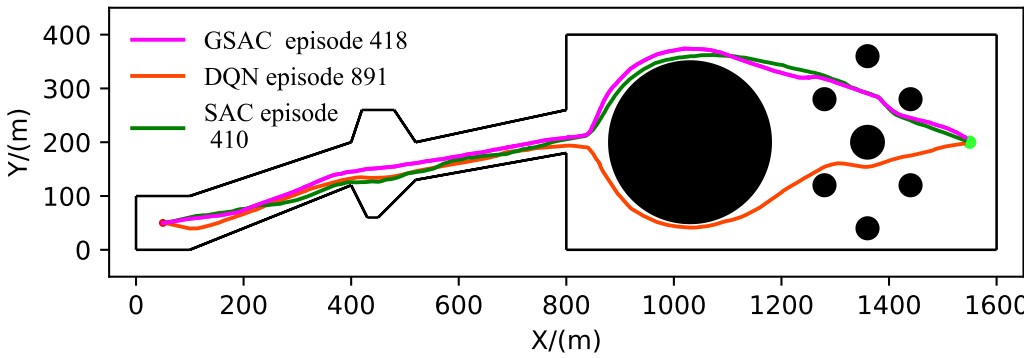

**Figure 10.** Static environment trajectory. ●: AUV; ●: Goal; ■: Obstacle.

### 4.2.2. Dynamic Environment Simulation

The test data of GSAC, SAC and DQN in static environment are shown in Table 5. Due to GSAC having a faster convergence speed in a dynamic environment, SAC and DQN are trained in more episodes to compare with GSAC.

The Success Episodes of GSAC at 501–1000 episodes is 496, which exceeds the test performance of SAC and DQN. The Truly Shortest Path Length of GSAC at 501–1000 episodes is 1631.30 m, which is 18.06 m less than that of SAC at convergence and 4.8 m more than that of DQN at 501–1000 episodes. The Average Length of GSAC in 501–1000 episodes is 1645.61 m, which is shorter than that of SAC and DQN. The Yaw Angle of GSAC at 501–1000 episodes is 4.92°, which is 0.06° more than that of SAC at converge and 0.74° more than that of DQN at 501–1000 episodes. At 1–1000 episodes, the difference in yaw angles between the three algorithms is negligible for path planning.

**Table 5.** Dynamic environment simulation result.

| Algorithm | Training Episode | Truly Shortest Path Length (m) | Average Length(m) | Yaw Angle (°) | Success Episodes |
|---|---|---|---|---|---|
| GSAC | 1–500 | 1630.04 | 1686.88 | 5.02 | 386 |
| GSAC | 501–1000 | 1631.30 | 1645.61 | 4.92 | 496 |
| SAC | 1–500 | 1642.20 | 1679.64 | 5.03 | 166 |
| SAC | 501–1000 | 1637.83 | 1695.90 | 4.88 | 223 |
| SAC | 1001–1500 | 1660.33 | 1776.35 | 5.25 | 198 |
| SAC | 1501–2000 | 1649.36 | 1693.83 | 4.86 | 468 |
| DQN | 1–500 | 1649.53 | 1735.75 | 4.20 | 381 |
| DQN | 501–1000 | 1626.50 | 1698.14 | 4.18 | 425 |
| DQN | 1001–1500 | 1641.92 | 1696.27 | 3.57 | 393 |
| DQN | 1501–2000 | 1623.21 | 1658.26 | 3.17 | 168 |

The curves of GSAC, SAC and DQN in the dynamic test environment are shown in Figure 11. The red axes in figure represent GSAC's test episodes, and the black axes represent test episodes of SAC and DQN. Figure 11a shows the task completion rate curves for the last 100 episodes. After 500 episodes of testing, GSAC can adapt to the environment and reach the goal stably, outperforming SAC and DQN. Figure 11b shows the average reward per step curves. The average reward curve of GSAC is smoother than that of SAC and DQN. Figure 11c shows the average yaw angle curves per step. The yaw angle of GSAC is smaller and more stable than that of SAC. Figure 11d shows the path length curves. The performance of GSAC in the 600-800 episodes outperforms the overall performance of SAC and DQN. GSAC has faster convergence, stable task performance, and shorter planned paths. The SAC and DQN curve have significant fluctuations. The optimal paths of GSAC, SAC and DQN in the dynamic test environment are shown in Figure 12. Compared with the SAC algorithm, GSAC can obtain more rewards in the dynamic underwater environment with faster convergence, stability, and robustness. Compared with the DQN algorithm, GSAC has faster convergence, stability, and robustness in the dynamic underwater environment.

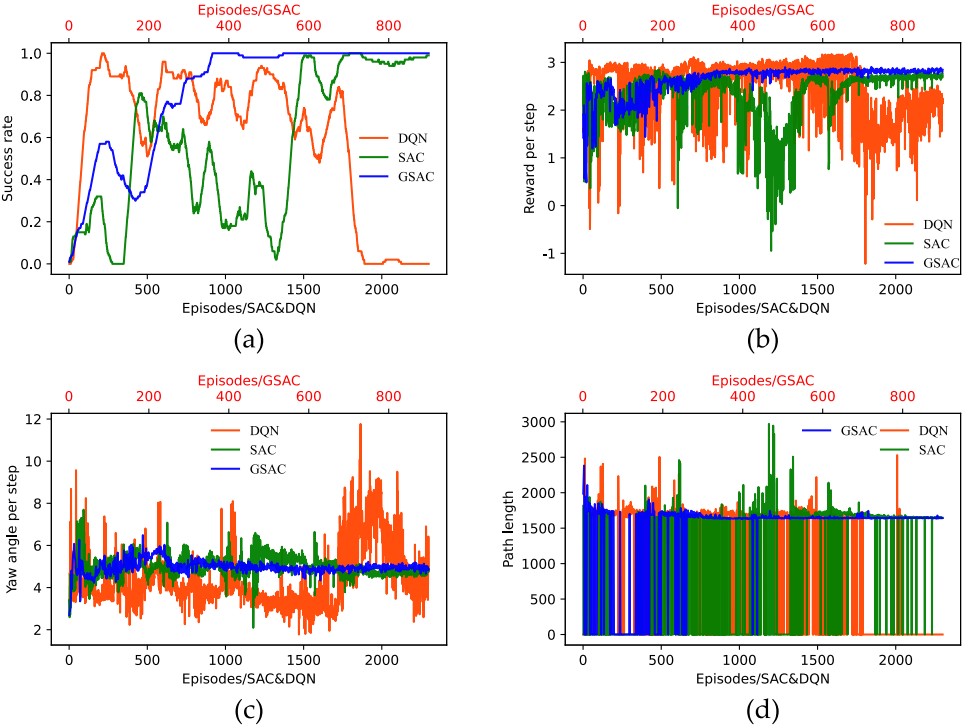

**Figure 11.** Dynamic environment simulation curves. (**a**) Success rate (last 100 episodes). (**b**) Reward per step. (**c**) Yaw angle. (**d**) Path length.

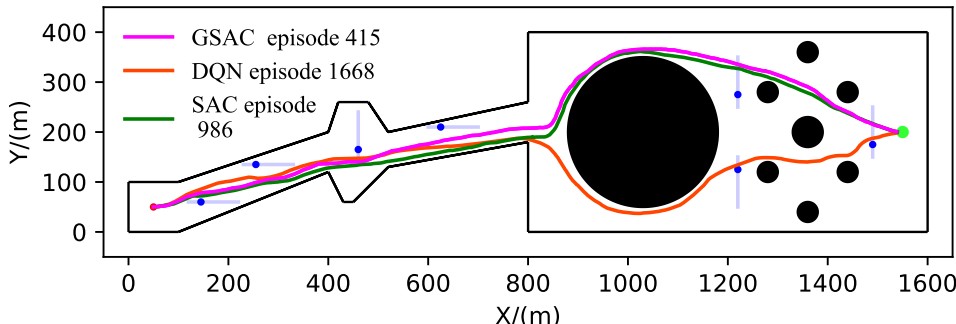

**Figure 12.** Dynamic environment trajectory. •: AUV; •: Goal; ■: Obstacle; •: Dynamic obstacle; —: Dynamic obstacle trajectory; —: Border.

## 5. Conclusions

To improve the adaptability of autonomous planning and the reliability of obstacle avoidance of AUVs in unknown underwater environments, the GSAC method is proposed in this paper. Considering the influence of the simulation environment, the obstacles in the simulation environment are regarded as agents and play a zero-sum game with the AUV. The zero-sum game problem is solved by improving the strategy of AUV and the obstacles, so that the simulation environment evolves intelligently with AUV path planning strategy. The proposed method increases the simulation environment's complexity and diversity, enables AUV to be trained in a variable environment specific to its strategies, and improves AUV's adaptability and convergence speed in unknown environments. Through simulation experiments in an unknown underwater simulation environment written in the Python language, the study verifies that GSAC can guide AUV to reach the target point in an unknown underwater environment, while avoiding large and small static obstacles (AUV sonar detection range is less than 1/6 of the large obstacle diameter), underwater canyons, and small dynamic obstacles in the simulated environment. In this study, modular reward rules are designed for multiple objectives of the AUV path planning problem to solve the sparse reward problem in AUV path planning. GSAC outperforms SAC and DQN in stability, convergence speed, and robustness.

**Author Contributions:** Conceptualization, Z.W. and H.L.; methodology, Z.W. and H.L.; software, Z.W. and H.L.; validation, Z.W., H.L. and Y.S.; formal analysis, Z.W., H.L. and Y.S.; investigation, H.Q. and Y.S.; resources, H.Q. and Z.W.; data curation, Z.W. and H.L.; writing—original draft preparation, H.L.; writing—review and editing, H.Q. and Z.W.; visualization, H.L.; supervision, H.Q. and Z.W.; project administration, H.Q. and Z.W.; funding acquisition, Z.W. All authors have read and agreed to the published version of the manuscript.

**Funding:** This research was funded by the China National Natural Science Foundation (grant number 52025111, 51979048, U21A20490, 51979057, 52131101).

**Institutional Review Board Statement:** Not applicable.

**Informed Consent Statement:** Not applicable.

**Data Availability Statement:** Not applicable.

**Conflicts of Interest:** The authors declare no conflict of interest.

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
