# Peer review of "Autonomous Underwater Vehicle Path Planning Method of Soft Actor–Critic Based on Game Training"

_jmse, doi:10.3390/jmse10122018_

Round 1
Reviewer 1 Report
Overall, I really like this paper. I particularly like that the test environments are more complex and more realistic than the training environment and how natural of an extension GSAC is on SAC. I've attached a few notes of things that really stood out to me as needing to be addressed but please don't interpret these as outweighing my overall positive impression.
One general note is that I think asking a local reader who wasn't involved in the work to review Section 3 in particular for clarity might be helpful as they could give iterative feedback in a way that isn't possible through this review system.

Author Response
Dear Reviewer:
Thank you very much for your comments and professional advices. These comments are all valuable and very helpful for revising and improving our paper, as well as the important guiding significance to our researches. We have studied comments carefully and have made correction which we hope meet with approval. Revised portion are marked in blue in the paper. According to the suggestion of reviewer 2, we have added deep Q-network to compare with GSAC and reviewed Figure 8-12.
The main corrections in the paper and the responds to the reviewer’s comments are as shown in the attachment.
Special thanks to you for your good comments.
Yours sincerely,
Hongde Qin
Corresponding author:
Name: Hongde Qin
E-mail: qinhongde@hrbeu.edu.cn

Reviewer 2 Report
1) Avoid using acronyms in the title. I mean "AUV" and "SAC". Also, I could not find the definition of SAC; what does it stand for?
2) Path planning and obstacle avoidance in autonomous systems (not necessarily underwater systems) is the subject of numerous publications. For instance, [R1]-[R3] propose algorithms for aerial and ground vehicles. Due to the similarity, the authors need to improve literature review and discuss these algorithm in the manuscript:
[R1]. "UAV Trajectory Planning in a Port Environment", 2020. [https://doi.org/10.3390/jmse8080592]
[R2]. "Constrained Control of UAVs in Geofencing Applications", 2018. [http://doi.org/10.1109/MED.2018.8443035]
[R3]. "A Hybrid Path Planning Method in Unmanned Air/Ground Vehicle (UAV/UGV) Cooperative Systems", 2016. [http://doi.org/10.1109/TVT.2016.2623666]
3) It is not clear from the text if the authors consider dynamical model for the vehicle or a static model? Is the vehicle modeled as a mass with controllable velocity?
4) Is the position of the obstacle known a priori? If so, how difficult is it to extend the proposed algorithm to account for dynamic obstacles?
5) Since weights \tau_i, I=1,...,5 in (10) can greatly impact the performance, the authors need to comment on how one can determine appropriate weights.
6) I am not sure if path length is a good index to compare GSAC with SAC. Also, is it possible to compare your results with a different state-of-the-art algorithm? GSAC is an upgraded version of SAC and it is expected to perform better than SAC. It would be nice to compare the performance of GSAC with a different algorithm.
Author Response
Dear Reviewer:
Thank you very much for your comments and professional advices. These comments are all valuable and very helpful for revising and improving our paper, as well as the important guiding significance to our researches. We have studied comments carefully and have made correction which we hope meet with approval. Revised portion are marked in blue in the paper. According to your suggestion, we have added deep Q-network to compare with GSAC and reviewed Figure 8-12.
The main corrections in the paper and the responds to the reviewer’s comments are as shown in the attachment.
Special thanks to you for your good comments.
Yours sincerely,
Hongde Qin
Corresponding author: Hongde Qin
E-mail: qinhongde@hrbeu.edu.cn

Round 2
Reviewer 2 Report
No further comment!